# Use of Traditional Mongolian Medicine in Children with Concussion

**DOI:** 10.3390/medicines10010005

**Published:** 2022-12-30

**Authors:** Orgilbayar Ganbat, Oyuntugs Byambasukh, Tserendagva Dalkh, Byambasuren Dagvajantsan

**Affiliations:** 1International School of Mongolian Medicine, Mongolian National University of Medical Sciences, Ulaanbaatar 14210, Mongolia; 2School of Nursing, Mongolian National University of Medical Sciences, Ulaanbaatar 13270, Mongolia; 3Department of Endocrinology, School of Medicine, Mongolian National University of Medical Sciences, Ulaanbaatar 14210, Mongolia; 4Department of Neurology, School of Medicine, Mongolian National University of Medical Sciences, Ulaanbaatar 13270, Mongolia

**Keywords:** concussion, Mongolian traditional medicine, folk medicine

## Abstract

(1) Background: There is no specific treatment for concussion in modern medicine, and existing treatment is only limited to resting and restoring cognition. For centuries, Mongolians have used traditional Mongolian medicine (TMM) methods to treat a variety of diseases such as *Baria zasal*. In this study, we aimed to explore the treatment parents and guardians seek when their children have suffered a concussion. (2) Methods: In this study, we used an online questionnaire. The study participants (*n* = 400) were randomly selected parents and guardians. The definition of *bariachi* is an advanced practitioner of *baria zasal*, which covers most of the massage therapy techniques mentioned in this study. (3) Results: In total, 72% of the parents and guardians went to a *bariachi* when their children suffered a concussion, while only 10.3% chose western medical hospitals. When asked what they did after the initial treatment was not effective, 47.8% of the participants responded that they went to the *bariachi*. Based on the days of treatment result, 11.8% reported on the beneficial effects of the treatment appearing in one day, and 60.3% in 1–3 days, which shows that the participants suffered a healing effect of the *baria zasal* shortly after application to their children. In the regression analysis, visiting a *Bariachi* was independent of age, gender, or even religion. (4) Conclusions: Although Western medicine is highly developed in Mongolia, the *baria zasal* of TMM has not lost its appeal in treating concussion. This suggests that *baria zasal* could be a unique method of concussion treatment even today. This also suggests that the techniques of *Baria zasal* should be further studied, and as in modern medicine.

## 1. Introduction

Concussions commonly occur not only in adults but also in children and adolescents [1]. Even though the symptoms of a concussion usually disappear in a short period of time [2,3], confusion and amnesia are often shown, especially in children [3]. There is a risk of experiencing post-concussion syndrome, resulting in concussion symptoms lasting weeks or even months instead of just a few days [4,5,6]. There is no specific treatment for concussion in modern medicine, and the existing treatment is only limited to rest and the recovery of cognition [7]. It is dehumanizing to merely observe a patient with symptoms that last for a short period of time [8]. It is important to introduce alternative methods to ease the patient’s distress. 

The Mongolian way of life and nomadic culture has existed for hundreds of years, and for centuries Mongolians have used traditional medicine methods, which are named traditional Mongolian medicine (TMM), to treat a variety of diseases [9,10,11]. TMM is well adapted to Mongolians’ nomadic lifestyle, as well as to Mongolia’s severe climatic conditions, culture, foods, etc. Treatments for concussion, bloodletting therapy, cupping therapy, and moxibustion therapies are still in everyday use to this day. Mongolian traditional medicine developed rapidly in the 7th, 12th, and 16th centuries, and the peak of its development was during the 16th and 17th centuries [9]. Otoch Sumbe Khamba Ishbaljir improved and updated the diagnosis and treatment of concussion, and we are using this treatment in our daily lives, but it is still not well studied [10]. In the historical and traditional medicine sutra, one of the most widely used methods of TMM is “*Baria zasal*”—mostly known as a massage treatment that is commonly used during concussion [9,12]. A *Bariachi* is an advanced practitioner of *baria zasal*. Most Bariachis are not formally trained, and they often practice *baria zasal* based on their instincts, natural talents, and abilities. Each *bariachi* uses different techniques to treat concussion because *bariachi* are not formally trained [9,12]. Accordingly, Mongolian scientists have stated that the method of treating concussion has still not been developed in a certain system, has remained highly empirical, and has not been introduced to neighboring counties [9,10,11,12]. 

TMM has been influenced by the socio-political situation until today. In 1921, after the Mongolian People’s Republic was declared, relations with Russia (at that time, the Soviet Union) intensified, which led to the introduction of modern medicine. From this moment on, as Western medicine spread, the use and development of TMM slowed down. After the transition to democracy in Mongolia in 1990, the practice of TMM was revived [13]. A study was done by J.A Bernstein in the 1990s, noting that Mongolians believe that *baria zasal* a more effective way for treating head trauma and headaches, rather than Western medicine [14]. In this study, we aimed to explore how this perception of Mongolians has changed compared to 20 years ago with the specific aim of studying which treatment parents and guardians seek when their children experience concussion. 

## 2. Materials and Methods

### 2.1. Data Source and Selecting Survey Participants

The study was conducted using the online questionnaire (Appendix A) because it was performed during the quarantine period of the COVID-19 pandemic in 2021. The study included the parents and guardians of children aged between 0 and 10. To reduce misapprehend factors that could influence the result of the study, people from all sectors, including civil servants, employees of private companies, etc., were selected as the target group. The survey link was sent to the staff of several organizations and private companies via email and posted on social media platforms. Prior to the online survey, responders were asked to read the consent form, and those who answered “yes” participated in the study. Participants had the choice to skip or refuse to answer some of the questions. In case they gave an incomplete answer, the results were calculated after excluding them from the survey. The design of the survey focuses on avoiding multiple responses, so that if the child had not experienced concussion before, there was no need to answer further questions. If the child had a concussion and was diagnosed with a brain injury by a diagnostic tool such as CT and MRI, responders were excluded (*n* = 29) from the study. In total, the data of 400 responders were analyzed in this study. The study was conducted according to the Helsinki Declaration, and it was approved by the medical ethics committee of the Mongolian National University of Medical Sciences (METc 2020/3-05).

### 2.2. Development of Online Questionnaires Survey

We followed several principles in developing and using an online questionnaire, to minimize bias [15]. For the participators’ convenience, the component of questions was designed with a user-friendly layout, enabling respondents from all ages who were familiar with using internet to participate. The format of the questionnaire was easy to read and fill out, with a short sentence structure. To avoid confusion, multiple-choice answers were not included. In addition, to make the questionnaire accessible to participants with low internet speed, we designed it in such a way that it would take as little time as possible to fill out. 

We conducted a pilot study of 20 people to measure the quality of the questionnaire in order to carry out internal validation. Then, we calculated the Cronbach alfa, which was 0.88.

In addition, to make it easier for participants to share their personal information, such as age and gender, the participants’ identification, such as their name, was not recorded, thus ensuring their privacy. 

### 2.3. Data Management and Statistical Analysis

We used an online survey platform in order to reduce and prevent the consolidation error to the result. We checked our results by comparing our analyses with the consolidated results as provided by the platform. The answers could not be altered by the researchers. Statistical analysis was performed by converting the data from the aggregated excel files to SPSS. The study characteristics were expressed as means with a standard deviation (SD) and as numbers with percentages in cases of categorical data. The frequency distributions of categorical variables were analyzed using the Pearson Chi-Square test. We analyzed which factors influenced people to choose bariachi after they experienced concussion, based on the logistic regression analysis. This included age, sex, education, religion, the cause of the concussion, and the reason for choosing bariachi. Exp (B) interval is reported with a 95% confidence interval (CI).

## 3. Results

The mean age of the survey respondents was 35.3 ± 9.1, of which 13.3% were male and 11.5% had a low level of education. In terms of religion, 59.6% were Buddhist, whereas 14.7% were other religious, including Christians, Muslims, or adherents to other religions. The remaining 25.8% were not religious (Table 1).

Of all the study participants, 78% of their children have been concussed. The most common symptoms reported were crying, vomiting, nausea, diarrhea, loss of appetite, and headaches. The terms of emotional changes after experiencing concussion were crying, anger, sadness, mood swings, and heightened emotion, represented in 66.3%, 23.8%, 6.3%, 5.5%, and 1.5%, respectively. In addition, 2% of the children experienced memory loss following a concussion and 4.8% observed a decrease in concentration in their children. Considering the causes of concussion, falls from heights were the most common cause of concussion in children at 67.8% (Figure 1). Other common causes were being hit in the head with something and stumbling, and others.

Most parents and guardians (72%) visited a bariachi as soon as their children suffered a concussion. As shown in Figure 2, 72% of the parents and guardians visited a *bariachi* when their children suffered from a concussion, while fewer than 10.3% chose western medical hospitals. As shown in Figure 2, other traditional Mongolian medicines include acupuncture and herbal medicine. Further treatments in this figure mean that the parents themselves take steps at home. When asked what they did after the initial treatment was not effective, 47.8% of the participants responded that they went to the *bariachi*.

Based on the days of the treatment result, 11.8% reported on the beneficial effects of the treatment appearing in one day, 60.3% in 1–3 days, 15.5% for 3–5 days, and 3% for 5–7 days. This shows that the participants experienced a healing effect of the *baria zasal* soon after it was applied to their children. 

Of the answers to the question “Why do you visit a *bariachi*?”, “By my own beliefs” and “based on previous experience” were possible predictors of visiting a *Bariachi* in the regression analysis (Table 2, *p* < 0.05) but not “advice by others”. In other words, when their child had suffered a concussion before and their condition improved consistently after a previous baria zasal, then the parents and guardians referred to a bariachi more often. Furthermore, female gender, a lower level of education, and religion were not associated with visiting a *Bariachi* (*p* > 0.05).

## 4. Discussion

We found that most parents and guardians consult a *bariachi* as their first choice if their child suffers from a concussion. Compared to the study conducted by J.A Bernstein 20 years ago [14], it shows that TMM still plays a major role in the treatment of a concussion, even after the rapid development of modern (Western) medicine in the intervening period. The study was conducted into the frequency with which Mongolians visit practitioners of Western medicine compared to practitioners of TMM when they have head injuries or headaches in the city of Darkhan, Mongolia. A total of 90 survey-based interviews were completed in the study [14]. Subjects were also asked to use a Likert-scale to rate the importance of factors in deciding whether to visit a traditional or a Western practitioner. The conclusion of the study was that visiting a *bariachi* was considered to be more effective in treating ‘head trauma’ or ‘headache’ among study participants. The fact that *baria zasal* is still widely used indicates that the way treatment is still considered beneficial by the population. There are several reasons why people choose to visit a *bariachi*: (a) It can be because it is less expensive and is easy to reach; (b) the patient’s symptoms disappear, and the patient feels better in a short time after getting massage treatment. Alleviating concussion symptoms through Baria’s zeal could be explained by a decrease in muscle tone, while arterial and venous blood flow improves during Baria zasal. As a result, serotonin increases, cortisol levels decrease, and parasympathetic activity is activated [16,17]. Furthermore, performing baria zasal on the head during a concussion improves certain parts of the body such as bioactive points (acupuncture points), which are activated to relax tense muscles. Furthermore, the neck range of motion and blood flow increase; pain decreases; the patient becomes more relaxed; and mental activity and the ability to bear intellectual stress are improved [9,10,11,12].

When we performed a regression analysis to establish the causes to the visit to *bariachi*, there were no specific ones. However, when children experience multiple concussion symptoms at the same time, they always visit the *bariachi*. Furthermore, their choice to visit a bariachi for the treatment of concussion instead of western medication, or any other therapy, is independent of age, gender, or even religion. *Bariachis* sometimes utilize religious elements such as *Dom zasal* in their *Baria zasal* [12]. For this reason, the *Baria zasal* in adults might be related to placebo effects [16]. Furthermore, it might be a reason for a lack of research into the effect of TMM treatments. Due to the lack of a theory that fully explains the effect of TMM, modern medicine tends to explain that traditional medicine treatment in terms of a patient’s trust in doctors and the placebo effect [16]. Therefore, in this study, we focused on children to find out how effective *baria zasal* is in treating children who have concussion. Although there is a strong religious affiliation in the TMM, the results of our study show that the *bariachi* visit was not connected with the religions of the people.

Concussion rates vary across countries and are dependent on the age of the child. We found a higher rate of 78% of our respondents who reported that their child had a concussion between the ages of 0 and 10 [1]. This indicates that concussion is very common in Mongolia. Although car accidents are very common in Mongolia, the causes of concussion in children are more likely to be domestic accidents, which are less likely to result in a hospital visit [14]. Within the category of domestic accidents, falls from a height make up the majority, with 67.8%; followed by hits on the head with an object, at 17.3%; and stumbling while walking, at 13.5%. 

According to the directives of western medicine, for the observation of symptoms without specific medication, it is not appropriate to keep a child without treatment even for a short period of time. In our study, the majority of the respondents observed that the symptoms showed up completely in 1–3 days. It can also be more appropriate to treat concussion with *baria zasal* than to treat it with symptomatic medicines. We found that of those who choose to take western medicine, half returned to the *bariachi*. Some studies have shown that people who receive the standard treatment of concussion to improve their physical condition and are discharged from hospital continuously experience discomfort for up to 1 month and end up visiting the hospital again [17]. The main part of the *baria zasal* is massage, and our research shows that concussion can be cured in a few days with *baria zasal*, although there is not enough research evidence to support the treatment of concussion, but massage might be effective. Only a few studies have found massage therapy to be effective at reducing concussion symptoms [18,19,20]. Therefore, further studies on the effect of *baria zasal* are needed.

We also aimed to obtain the attention of researchers and to introduce them to the treatment methods of TMM, which are not well known worldwide. Even though this method of treating concussion has been applied by Mongolians for a long time [9,10,11], This time we only purposed to show the acquainted way of treating this issue. For future research, we aim to adopt a more scientific approach to studying concussion by using modern technology to study the efficacy of *bariachi* in relation to its outcomes.

The main limitation of the present study is the sampling method. An online survey is not a suitable tool to reach elderly respondents who might be inexperienced with internet usage; however, our research did not directly target older people. Another limitation is that we were not entirely certain about including children who had had a confirmed concussion. Even the diagnosis of a concussion in Western medicine is complicated. Therefore, we asked in the questionnaire whether there was any radiological alteration in the brain at the time of the concussion. Those who had experienced such an alteration were subsequently excluded from the study.

## 5. Conclusions

The majority of parents and guardians visited *bariachi* from the time their children suffered a concussion in Mongolia. Their choice to visit a *bariachi* for concussion treatment in comparison to any other therapy was independent of their age, gender, or even religion. These factors suggest that the *baria zasal* of TMM has not lost its appeal in treating concussion, while Western medicine is gradually developing in Mongolia.

## Figures and Tables

**Figure 1 medicines-10-00005-f001:**
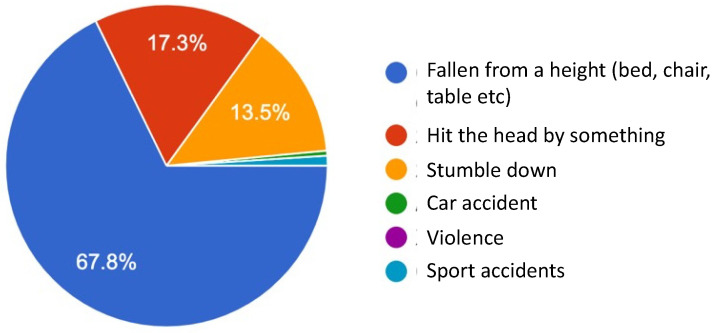
Causes of concussion in children.

**Figure 2 medicines-10-00005-f002:**
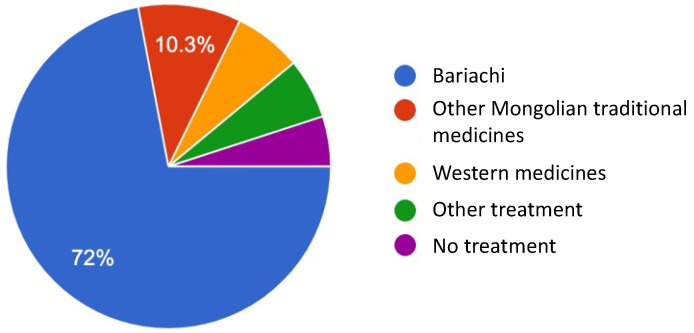
The treatments that parents and guardians sought when their children experienced concussion.

**Table 1 medicines-10-00005-t001:** General characteristics of the parents and guardians.

Variable	Total Population (*n* = 400)
Mean age ± SD, year	35.3 ± 9.1
Sex, *n* (%)	
Male	53 (13.3)
Female	347 (86.3)
Education, *n* (%)	
Low, medium	46 (11.5)
High	354 (88.5)
Religion, *n* (%)	
Buddhist	186 (59.6)
Christian	4 (1.3)
Islamic	2 (0.6)
Shamanist	40 (12.8)
No religion	80 (25.8)

Data are presented as mean ± SD and number (percentages, %).

**Table 2 medicines-10-00005-t002:** Possible predictors of visiting a *Bariachi*.

Variable	Exp (B) Interval	95% Confidence Interval	*p*-Value
Age: <35/≥35	1.24	0.70–2.21	0.457
Gender: male/female	0.49	0.18–1.31	0.159
Education: high/low	0.93	0.40–2.13	0.859
Religion: No/Yes	0.83	0.48–1.42	0.499
Number of symptoms: <5/≥5	1.74	0.91–3.33	0.091
Causes of concussion			
Sports accidents	1.00	-	-
Fallen from a height	1.10	0.54–2.26	0.776
Hit the head by something	0.90	0.44–1.84	0.776
Stumble down	0.58	0.28–1.20	0.149
Reason to visit a *Bariachi*?			
No reason	1.00	-	-
Advice by others	0.90	0.46–1.79	0.784
By my own beliefs	2.11	1.23–2.63	0.007
Based on the previous experience	2.36	1.38–4.03	0.002

Data are presented with Exp (B) with 95% confidence interval.

## Data Availability

The data used to support the findings of this study are available from the corresponding author upon request.

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
