# Peer review of "Use of Traditional Mongolian Medicine in Children with Concussion"

_medicines, 2022, doi:10.3390/medicines10010005_

Round 1

Reviewer 1 Report

The authors have attempted to study the effectiveness of the Traditional Mangolian Medicine technique 'baria zasal' in the treatment of concussion in children. The attempt made by the authors to give scientific validation to the traditional system of medicine is appreciable. However, some of the concerns regarding the study have to be answered.

  1. Has a scientific determination of concussion been done ?
  2. Please provide the questionnaire either as main file or supplementary file.

Author Response

The authors have attempted to study the effectiveness of the Traditional Mongolian Medicine technique 'Baria social' in the treatment of concussion in children. The attempt made by the authors to give scientific validation to the traditional system of medicine is appreciable. However, some of the concerns regarding the study have to be answered.

  1. Has a scientific determination of concussion been done ?

Response: It was a limitation to our study. Because we performed a retrospective study based on the questionnaire. The following sentences have been added to the limitation section:

Another limitation is that we are not entirely certain about including children who have had a confirmed concussion. But even the diagnosis of a concussion in Western medicine is complicated. Therefore, we asked in the questionnaire whether there was any radiological alteration in the brain at the time of the concussion. They were subsequently excluded from the study.

***.

  1. Please provide the questionnaire either as main file or supplementary file.

Response: As per your suggestion, the survey form is attached.

***.

To the editors and all reviewers: thank you all for your constructive comments and suggestions. Your guidance has helped to improve our manuscript.

Reviewer 2 Report

on line 102 ...... you need to explain briefly what the value of validity and reliability is from the questionnaire and its meaning is explained clearly even though your questionnaire is in the form of an online questionnaire

on line 179-180 .... you need explain what the benefit of a bariachi/baria zasal and make connection what the reliathionship with is and provide the more evidance base practice and cite  it. 

Author Response

on line 102 ...... you need to explain briefly what the value of validity and reliability is from the questionnaire and its meaning is explained clearly even though your questionnaire is in the form of an online questionnaire.

Response: We have rewritten this section as follows.

... We conducted a pilot study of 20 people to measure the quality of the questionnaire in order to carry out internal validation. Then we calculated Cronbach alfa that was 0.88.

***.

on line 179-180 .... you need explain what the benefit of a bariachi/baria zasal and make connection what the reliathionship with is and provide the more evidance base practice and cite  it. 

Response: Based on your suggestion, the following section has been added to the revised version.

.... Alleviating concussion symptoms through Baria's zeal could be explained by a decrease in muscle tone, while arterial and venous blood flow improves during Baria zasal. As a result, serotonin increases, cortisol levels decrease, and parasympathetic activity is activated [16-17]. Furthermore, by performing baria zasal on the head during a concussion, there are some improvements shown in the body such as bioactive points (acupuncture points) activate to relax tense muscles, the neck range of motion and blood flow increase, and pain decreases, also relaxation, mental activity, and the ability to bear intellectual stress are improved [9-12].

***.

To the editors and all reviewers: thank you all for your constructive comments and suggestions. Your guidance has helped to improve our manuscript.

Reviewer 3 Report

Very interesting study! Below are some comments on this manuscript:

1. Headache is the most commonly reported symptom of concussion, instead of confusion and amnesia in children?

2.More accurate meaning of "Nowadays, ...inhumane"?

3. It seems that there are inclusion criteria for concussion children and adolescent, as well as participant inclusion consent form, is it appropriate to state that "no special requirement to include participant in this study"?

4. Please check up the tenses through the whole manuscript, and make sure to use the past tense in the research methods and conclusion sections.

5. Based on Table 1, it was only 1.3% Christians were included in this study, is it accurate to record 14.7% in the text?

6. According to study results, visit a bariachi for concussion is just dependent of participants' belief and previous experience, is it scientific to draw a conclusion that this method might be unique way for concussion treating?

Author Response

Very interesting study! Below are some comments on this manuscript:

  1. Headache is the most commonly reported symptom of concussion, instead of confusion and amnesia in children?

Response: We assume that the absence of these symptoms may be due to the fact that the symptoms have disappeared because parents immediately consult Bariachi after a concussion.

***.

2.More accurate meaning of "Nowadays, ...inhumane"?

Response: We have rewritten this section as follows.

... It is dehumanizing to merely observe a patient with symptoms that last for a short period of time [8]. It is important to introduce alternative methods to ease the distress.

***.

  1. It seems that there are inclusion criteria for concussion children and adolescent, as well as participant inclusion consent form, is it appropriate to state that "no special requirement to include participant in this study"?

Response: We have removed and revised these sentences.

***.

  1. Please check up the tenses through the whole manuscript, and make sure to use the past tense in the research methods and conclusion sections.

Response: The manuscript was edited by English-speaking experts.

***.

  1. Based on Table 1, it was only 1.3% Christians were included in this study, is it accurate to record 14.7% in the text?

Response: We apologize for the miswriting. In fact, the 14.7% concerned all other religions, including Christians, Muslims, and followers of other religions. We have revised the sentence.

In terms of religion, 59.6% were Buddhist whereas 14.7% were other religious, including Christians, Muslims, or adherents to other religions. The remaining 25.8% were not religious (Table 1).

***.

  1. According to study results, visit a bariachi for concussion is just dependent of participants' belief and previous experience, is it scientific to draw a conclusion that this method might be unique way for concussion treating?

Response: We have deleted the sentence in the conclusion.

***.

To the editors and all reviewers: thank you all for your constructive comments and suggestions. Your guidance has helped to improve our manuscript.

Round 2

Reviewer 1 Report

The authors have made considerable changes to the manuscript by incorporating responses to the reviewer's comments.  The study shows the most prevalent TMM method of treatment for concussion. However, further studies are suggested in future to determine the efficacy of baria zasal in treating concussion.

Reviewer 3 Report

Accept!